# EfficientPNet—An Optimized and Efficient Deep Learning Approach for Classifying Disease of Potato Plant Leaves

**Tahira Nazir [1], Muhammad Munwar Iqbal [2], Sohail Jabbar [3], Ayyaz Hussain [4] and Mubarak Albathan [3,*]**

1.  Faculty of Computing, Riphah International University, Gulberg Greens Campus Islamabad, Islamabad 45320, Pakistan; tahira.nazir@riphah.edu.pk
2.  Department of Computer Science, University of Engineering and Technology, Taxila 47080, Pakistan; munwar.iq@uettaxila.edu.pk
3.  College of Computer and Information Sciences, Imam Mohammad Ibn Saud Islamic University (IMSIU), Riyadh 11432, Saudi Arabia; sjjabar@imamu.edu.sa
4.  Department of Computer Science, Quaid-i-Azam University, Islamabad 44000, Pakistan; ayyaz.hussain@qau.edu.pk
*   Correspondence: mmalbathan@imamu.edu.sa; Tel.: +966-503-451-575

**Abstract:** The potato plant is amongst the most significant vegetable crops farmed worldwide. The output of potato crop production is significantly reduced by various leaf diseases, which poses a danger to the world's agricultural production in terms of both volume and quality. The two most destructive foliar infections for potato plants are early and late blight triggered by *Alternaria solani* and *Phytophthora infestans*. In actuality, farm owners predict these problems by focusing primarily on the alteration in the color of the potato leaves, which is typically problematic owing to uncertainty and significant time commitment. In these circumstances, it is vital to develop computer-aided techniques that automatically identify these disorders quickly and reliably, even in their early stages. This paper aims to provide an effective solution to recognize the various types of potato diseases by presenting a deep learning (DL) approach called EfficientPNet. More specifically, we introduce an end-to-end training-oriented approach by using the EfficientNet-V2 network to recognize various potato leaf disorders. A spatial-channel attention method is introduced to concentrate on the damaged areas and enhance the approach's recognition ability to effectively identify numerous infections. To address the problem of class-imbalanced samples and to improve network generalization ability, the EANet model is tuned using transfer learning, and dense layers are added at the end of the model structure to enhance the feature selection power of the model. The model is tested on an open and challenging dataset called PlantVillage, containing images taken in diverse and complicated background conditions, including various lightning conditions and the different color changes in leaves. The model obtains an accuracy of 98.12% on the task of classifying various potato plant leaf diseases such as late blight, early blight, and healthy leaves in 10,800 images. We have confirmed through the performed experiments that our approach is effective for potato plant leaf disease classification and can robustly tackle distorted samples. Hence, farmers can save money and harvest by using the EfficientPNet tool.

**Keywords:** agriculture; classification; deep learning; transfer learning; convolutional neural networks; EfficientNet; potato diseases





## 1. Introduction

According to the UN Food and Agriculture Organization (FAO), the global population could reach 9.1 billion by 2050. Due to the rising population, food consumption will increase [1]. In the meantime, the lack of farmland and access to clean water makes it hard for nutrient levels to rise. In order to meet human needs, there is an immediate need to boost food security while using the least amount of growing area. As opposed to this, a number of crop anomalies cause a significant decrease in meal productivity and quality. Therefore,

immediate detection of these plant pathogens is necessary, as they have the potential to reduce agricultural profits and increase inflation rates. Such outcomes may cause market-wide economic uncertainty. Additionally, agricultural crop disorders at their most severe stages can wipe out harvests and cause hunger in a country, especially in developing nations with poor incomes. Typically, plant assessments are performed with the aid of domain specialists; however, this is a laborious and time-consuming task that depends on the participation of local professionals. Additionally, such methods of crop evaluation are not regarded as highly trustworthy, and it is difficult for people to individually evaluate each crop [2]. Therefore, it is critical to accurately and promptly identify the numerous plant illnesses that might prevent growers from deploying pricey treatment techniques while improving the food growth rate. The science world is concentrating its effort on the creation of computerized plant illness diagnosis and recognition systems to address the aforementioned issues with manual approaches [3].

Despite the existence of numerous different crops, such as tomatoes, onions, strawberries, and cherries, among others, the potato plant is a highly consumed crop around the globe. The potato crop is regarded as the major staple by more than a billion people globally, and is considered the third largest food crop on the planet after rice and wheat. More than 300,000 tons are produced globally each year, providing both nutrients and an essential source of calories for people [4]. In addition to providing a sizeable share of the world's nutrition, potatoes are a common source of raw ingredients for industry. Potatoes are produced all over the world, with the top three exporters being China, India, and Russia [5].

Following a survey performed by the UN Food and Agriculture Organization (FOA), the prevalence of many illnesses, the majority of those which originate from the leaves of the potato crop and cause a reduction in output amount from 9% to 11% annually [6], is the main obstacle to the pace of potato growth. To examine potato crop leaf disorders, the scientific world initially used methods from the fields of biological sciences and cell biology [7,8]. These methods, however, exhibit high processing complexity and demand a significant need for expert skills [9]. The majority of agricultural production is done by low-income individuals; hence, such pricey methods are not practical for farmers [10]. The rapid advancement of machine vision and object classification algorithms is being used in existing works to design automated methods for identifying crop pathogens. Image processing and machine learning (ML) studies are receiving more focus, and these methods are emerging as appealing alternatives to ongoing crop infection surveillance. Several conventional ML predictors, such as K-Nearest Neighbors (KNN), Random Forest Tree (RFT) [11], and Support Vector Machine (SVM), are highly employed in existing works for accomplishing classification tasks related to various plant-related diseases. Although these ML techniques are simpler to comprehend and only need a minimal quantity of samples to build models, they take time and rely heavily on expert human capital. Additionally, the classic ML information computation methods consistently necessitate a compromise between processing effort and classification results [12].

Deep learning (DL) techniques are currently being evaluated to address the shortcomings of ML algorithms. Different DL methodologies, including CNN [13], RNN [14], and long short-term memory (LSTM) [15], are currently widely praised in the field of food security. DL methods are capable of accurately estimating the informative collection of sample feature characteristics without the assistance of domain experts. Both these strategies for object recognition and deep learning (DL) imitate how the human brain functions when a person locates and recognizes a variety of items by looking at examples of them. DL approaches provide reliable results in the field of modern agriculture research, and are effectively suited to a variety of jobs, whereas different kinds of deep neural networks (DNNs) exhibit greater precision than multispectral evaluation. The agricultural production field is intensively investigating methods such as GoogLeNet [16], DenseNet [17], Inception, VGG [18], and Residual Net [19] for problems including quantifying grain volume, detecting plant heads, quantifying fruits, crop disorder diagnosis and categorization, etc.

Because of their capacity to utilize the structural and morphological information afrom the investigated images, these approaches are able to demonstrate excellent recognition accuracy while minimizing processing effort [20].

Even though experts have carried out a significant amount of work to classify potato crop leaf infections, it remains difficult to identify illness in the initial stages, as infected and healthy plant sections share many similar characteristics [20]. Recognition is made more difficult by varied plant leaf shapes, fluctuations in lighting and luminosity, the inclusion of distortion, and blurring in the processed images. Thus, there remains an opportunity for potential improvement in terms of computing power as well as correctness in identifying potato plant diseases. In the presented work, we attempted to tackle the existing problem of potato plant leaf disease classification by proposing an effective DL approach, namely, EfficientPNet. We have modified the existing EfficientNet-v2 model by introducing an attention mechanism (AM) and additional dense layers at the end of the framework structure. The presented EfficientPNet approach robustly extracts high-level signs of infected regions and associates them with related groups via employing an end-to-end training mechanism. In addition, the AM strategy improves the recall power of the proposed solution by passing relevant details of noticeable attributes such as diseased areas of plant leaves. The distinctive contributions of this work can be elaborated as follows:

(1) An effective light DL approach called EfficientPNet is suggested that is proficient in calculating relevant and distinctive sample characteristics and shows improved potato plant leaf disease classification results with little computational effort.

(2) The model includes the pixel and channel attention approach in the feature computation phase, which improves its ability to comprehend crosslinks and spacewise orientation properties to accelerate the diagnosis of potato leaf disorders in realistic scenarios.

(3) Transfer learning and multi-class focal loss are adopted to cope with the problem of class imbalance and network overfitting, which improves the precision of classifying potato leaf infected regions.

(4) In order to demonstrate the effectiveness of the suggested EfficientPNet model, we performed huge comparison evaluations to check the classification results by utilizing a collection of images of potato crop disease taken from a standard sample repository called PlantVillage. The suggested method successfully categorizes potato crop illnesses, even in the context of challenging external factors such as noise, distortion, unbalanced lighting, and variations in the shape, color, and placement of infection marks.

(5) To increase the size and ensure balance between the training and testing datasets, we have applied data augmentation techniques. Using these data augmentation techniques, the classifier become more able to generalize.

The rest of this paper is organized as follows: related works is presented in Section 2, and the proposed method in Section 3; we discuss the obtained results in Section 4; finally, the work is concluded, and our future research plans are elaborated in Section 5.

## 2. Related Works

There have been several approaches presented for potato leaf disease detection from leaf images. In [21], the authors proposed a pre-trained ResNet50 CNN model for the classification and detection of plant diseases. This method was applied to potato leaves taken from the PlantVillage dataset. The presented approach included augmentation and segmentation, which were then passed to ResNet-50 for classification, achieving 98% accuracy. The method performed well; however, its accuracy depends on augmentation and needs further improvements. Bhagat et al. [22] presented bag-of-words (BoWs) and SURF-based techniques for the identification of potato leaf diseases. The bag of words approach was utilized for feature extraction in the initial phase. After that, the SURF method was selected to extract the strongest features, which were then passed to an SVM for classification. Experiments were performed on potato leaves taken from the PlantVillage

dataset, and the model attained 97% accuracy. The method in [22] performed well; however, the model did not consider unseen or real-world samples.

Pal et al. [23] proposed the AgriDet (Agriculture Detection) approach. Their method utilized the conventional Inception-Visual Geometry Group Network and Kohonen for the detection and identification of potato leaf disease. The multi-variate Grabcut was applied to reduce the occlusion problem. This method was applied to the PlantVillage dataset to detect and segment potato leaf disease classification. The model achieved good results, with 92.12 % accuracy. The presented approach can tackle the overfitting problem through the dropout layer. However, it is unable to recognize multiple instances of the same disease in one image. Yu, H. et al. described an improved deep learning model for classifying potato plant leaf diseases in their paper [24]. They used a convolutional neural network (CNN) and a transfer learning approach to train their model on a large dataset of potato leaf images. The model achieves high accuracy rates in classifying different types of diseases, and outperforms several other deep-learning models in terms of accuracy and training time [24,25].

Chen, X. et al. presented a study on potato leaf disease classification using an improved deep learning model. The authors used a modified Inception-V3 model and a transfer learning approach to train the model on a dataset of potato leaf images. The model achieves high accuracy rates in classifying different types of diseases and outperforms several other deep-learning models in terms of accuracy and training time [26]. In [27], Kang et al. proposed a lightweight CNN-based approach for the recognition of potato leaf diseases. The authors utilized multi-scale pyramid fusion technology for efficient feature selection. This fusion of features was achieved using the improved backbone model and optimized features. This lightweight technique recognized and identified plant leaf diseases, achieving 93% accuracy. However, the presented model needs further improvements in accuracy.

To detect and classify potato leaves, Kumar et al. [28] presented an automated method based on Gaussian filtering and Fuzzy c-means clustering. This method extracted different types of features, including textual, geometrical, and statistical features. The extracted features were then passed to a PCA for efficient feature selection. At last, several classifiers were employed for the classification of potato leaves. The unbalanced data makes [29] machine learning models more biased and leads to overfitting issues. This study shows a way to add more information to data based on an image-to-image translation model. This helps eliminate the bias from adding these bad potato leaf images. To produce pictures representing more obvious disease textures, the authors suggested that the augmentation approach translates healthy and unhealthy leaf images and uses attention processes.

Rashid et al. [30] proposed a multi-level DL-based model for the recognition of potato leaf diseases. In the initial stage, the YOLOv5 technique was employed for the segmentation of images. Second, the Deep CNN model was utilized for potato leaf identification from images. Experiments were performed on a proprietary dataset and achieved good results. However, the presented model is unable to detect multiple diseases from a single image. Tiwari et al. [31] proposed a deep learning technique for the detection of potato leaf diseases. Their model was based on numerous approaches. In the first step, features were extracted through a VGG19 model. The extracted features were then classified using different classifiers, in which logistic regression performed well compared to the others, achieving 97.8% accuracy on the PlantVillage dataset. The presented model needs further improvements to efficiently detect unseen examples. Similarly, a CNN approach was utilized in [32] to recognize potato leaf diseases. The technique was based on the Adam optimizer and cross-entropy for model analysis. The final classification was performed using a softmax layer. Another CNN-based approach was employed in [33] for the detection of potato leaf diseases. Experimentation was performed on the Kaggle dataset, and the model attained 97% accuracy. However, the presented model tackles only binary classification. Iqbal et al. [34] proposed a method for the segmentation and classification of potato leaf diseases. The PlantVillage dataset was utilized for evaluation of the proposed technique. The random forest approach was employed for classification of leaves into two types,

diseased or healthy, with an accuracy of 97%. A deep learning technique was proposed to efficiently detect potato leaf diseases using PlantVillage dataset in [35]. The model was based on the lightweight MobileNet-V2, which was then modified using the addition of a layer in the model. The model achieved 97.33% accuracy on the classification of potato leaf disease. However, this model is computationally light in terms of time. In [36], the authors presented a deep learning-based approach for the classification of potato leaf diseases. The proposed technique was based on four types of models: MobileNet, VGG16, VGG19, and ResNet. Fine-tuning of parameters was performed to enhance the accuracy of the proposed model. Experiments were performed on the PlantVillage dataset, achieving 97.8% accuracy. However, the presented approach did not tackle real-world samples.

## 3. Materials and Methods

Our proposed work is based on the EfficientNet approach called improved EfficientNetV2 for the recognition and classification of potato leaf diseases. To test and validate the performance of the proposed system, the PlantVillage dataset, with a total number of 54,306 images of potato plants, was utilized. To balance this dataset in each class, we applied data augmentation techniques. The proposed work is focused on improving the EfficientNet approach for potato leaf disease recognition and classification by introducing additional layers at the bottom of the model. These additional layers were designed to enhance the model's performance by allowing it to identify more complex patterns and features in the images. The improved model, called improved EfficientNetV2, was trained on a large dataset of potato leaf images consisting of both healthy and diseased leaves. The model was trained using a supervised learning approach in which it was provided with labelled examples of healthy and diseased leaves and learned to classify new images based on the patterns identified in the training data. The additional layers at the bottom of the improved EfficientNetV2 model allow it to capture more low-level features and patterns in the images, which in turn can improve the accuracy and robustness of the model. Techniques such as transfer learning, data augmentation, and regularization can be employed to further improve the model's performance.

The proposed work has the potential to contribute to the development of more accurate and reliable models for potato leaf disease recognition and classification, which can help farmers and agricultural researchers in their efforts to improve crop yields and reduce losses. The improved model has additional layers at the bottom of the model, which help to enhance the performance. The complete flow of our improved model is shown in Figure 1. The overall process is explained in Algorithm 1.

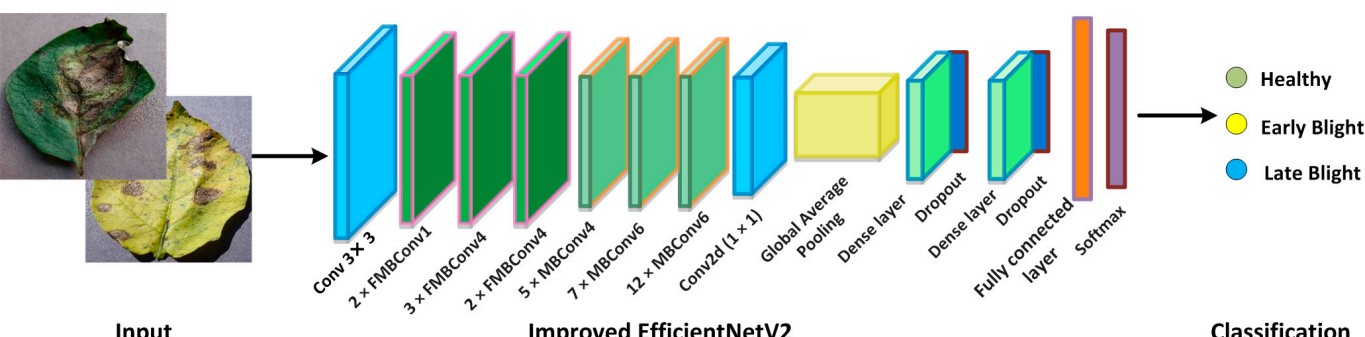

**Figure 1.** Flow of the proposed EfficientNetV2 framework.

---

**Algorithm 1:** Steps followed by EfficientPNet for potato plant leaf abnormality categorization

---

START
INPUT: TP, Labels
OUTPUT: The category of potato plant leaf diseased region, EfficientPNet
        TP: Total potato images with various abnormalities.
        Labels: Class of each potato sample
        EfficientPNet: improved EfficientNet-V2 model.
//Data preparation and augmentation to balance dataset
        Data Augmentation (x)
         SampleDimension $\leftarrow$ [ j h]
         // Labels associated with each input sample
         $\ddot{A} \leftarrow$ ReadClassLabel (TP, Class)
// training phase //Functions

1.    EffiNetV2(): employed to measure the keypoints with EfficientNet-V2 network
2.    EvaluatFramework(): employed to accomplish the model training

        // Improved EfficientNet-V2 model
EfficientPNET $\leftarrow$ EffiNetV2 (SampleDimension, $\ddot{A}$)
        [ TrainingPart, TestPart $\leftarrow$ Database distribution
        For each sample c in $\rightarrow$ TrainingPart
           Compute *improved-EfficientNet-V2* features $\rightarrow tm$
        End
        Utilize *tm* images EfficientPNet training, and calculate time
        £abelA $\leftarrow$ IdentifyPotatoLeafAffectedAreaLabel (*tm*)
        *Ap* $\leftarrow$ EvaluatFramework (*improved-EfficientNet-V2*, *LoclizeA*)
// test phase
        For each image *C* in $\rightarrow$ TestPart

          (a)    $\beta C \leftarrow$ Compute features via employing the trained model EfficientPNet
          (b)    [*ConfidenceScore*, *ClassLabel*] $\leftarrow$ Predict ($\beta C$)
          (c)    show samples *ClassLabel*

          End
   Exit

---

### 3.1. EfficientPNet Framework

A robust set of image features is essential to obtaining superior classification results, as it directly helps to distinguish the numerous image data groupings. The use of dense DL networks can help in calculating a collection of more effective characteristics, which in turn causes the recall rate of methods to increase [16]. The deployment of these CNN techniques depends heavily on the availability of processing power and memory needs, which places a computational constraint on the models when deep networks are used. Consequently, the cost of computing and the results of the evaluation are always tradeoffs. For this reason, it is necessary to provide a system for identifying leaf diseases that can demonstrate improved classification accuracy while maintaining computing costs [37]. In this study, we introduce a simple and reliable computational strategy to improve model performance for categorizing various anomalies.

An enhanced EfficientNetV2-B4 model is introduced for the identification of potato plant diseases and given the name of EfficientPNet. EfficientNetV2is an expanded version of EfficientNet [38]. Essentially, the improved EfficientNetV2 model is presented to increase available resources while maintaining a high recall rate. The improved EfficientNetV2 model was created using a quick and effective composite scaling method that enables a regular ConvNet to be scaled to any resource limitations while maintaining the method capability. Therefore, the proposed approach offers an ideal choice for network design, i.e., network layers or feature vector size, as well as an optimal solution for computing cost. The EfficientNetV2 technique conducts classification operations robustly and only uses a limited number of model parameters. Furthermore, it performs well in terms of

efficiency compared to other methods such as GoogleNet [16], AlexNet [39], DenseNet [40], ResNet [41], and MobileNet [42].

The motivation behind EfficientNet-V2 with dense layers for recognition of potato leaf diseases is that it is an efficient and lightweight approach that requires less training time and contains fewer parameters. The EfficientNetV2 approach makes use of neural architecture search to increase classification accuracy while reducing the size of feature vectors and training time (NAS). Additionally, by including the Fused-MBConv (FMBConv) blocks [43] in the EfficiceiNetV2 architecture, the operative power is optimized and mobile or server accelerators are employed effectively, whereas the conventional EfficientNet technique, which only uses depth-wise convolutions, uses MBConv blocks [44] as its primary building block. Despite the fact that depth-wise convolutions reduce the number of operations required, they do not fully utilize new hardware accelerators. The EfficientNetV2 technique fully utilizes both MBConv and FMBConv blocks to achieve computational gains. The depth-wise 3*3 convolution is replaced in the FMBConv by conventional 3*3 convolution layers. The main objective is to boost the implementation speed of the model while keeping the classification results [45] as shown in Figure 2.

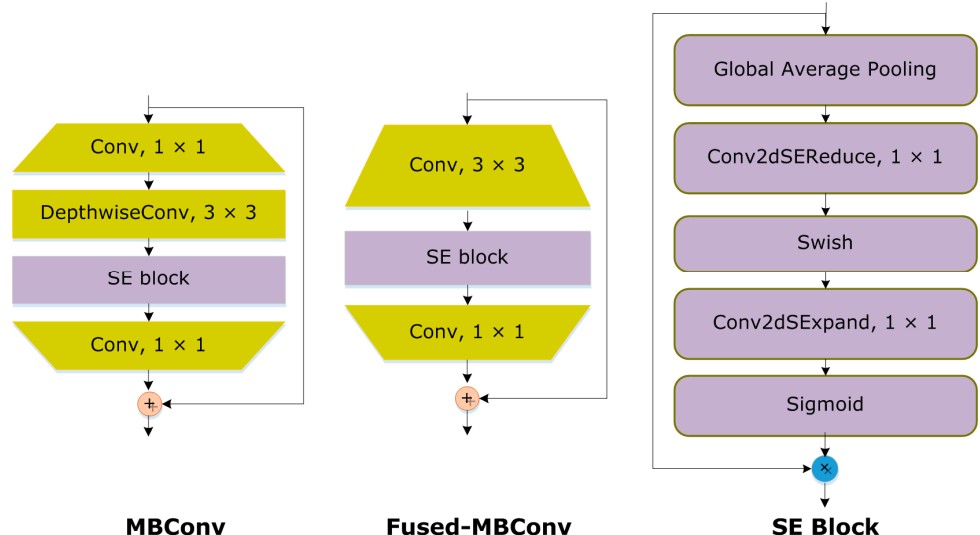

**Figure 2.** Graphic form of MBConv4, Fused-MBConv4, and SE block.

We used EfficientNetV2 with the B4 architecture to complete the classification task. The B4 base was chosen primarily because it shows a good trade-off between time complexity and model classification performance. Table 1 provides a thorough overview of the enhanced EfficientNetV2 model. The revised EfficientNet-V2 model uses FMBConv blocks at the bottom layers while using MBConv blocks with 3*3 and 5*5 convolutions, squeeze-and-excitation block (SEB) [46], and swish activation at the advanced level. The MBConv blocks preserve an up-set residual link through the SEB to produce robust classification results.

ReLU activation (ReLUAF) is replaced in the framework by the swish activation function (*SAF*) [47], as ReLU excludes values lower than zero and loses an essential component of the used ECG signal. The following equation can be used to calculate the *SAF* (1):

$$SAF(x) = X.Sigmoid(x) \tag{1}$$

Additionally, a Batch normalization layer was added at the beginning of a framework to down-sample the input image sizes. Only three FMBConv blocks were used, as they include many parameters for large values of O. After the MBConv, a global average pooling layer was introduced to reduce the model parameters in order to prevent the issue of model overfitting. Together with the ReLUAF and dropout layers, we included two additional inner-dense layers that help to compute the more effective collection of image

characteristics by effectively presenting them. A dropout rate of 30% was chosen arbitrarily in order to progress the model's performance. At the end, a softmax layer was utilized for the classification of potato leaf diseases.

**Table 1.** Details of blocks and layers used in the proposed model.

| Sr No. | Layers |
| --- | --- |
| 1 | BatchNormalization |
| 2 | ConvL (3 × 3) |
| 3 | 2 × FMBConv1 Block |
| 4 | 3 × FMBConv4 Block |
| 5 | 2 × FMBConv4 Block |
| 6 | 5 × MBConv4 Block |
| 7 | 7 × MBConv6 Block |
| 8 | 12 × MBConv6 Block |
| 9 | Conv2d (1 × 1) Block |
| 10 | Global average pooling |
| 11 | Dense Layer |
| 12 | Dropout |
| 13 | Dense Layer |
| 14 | Dropout |
| 15 | FC Layer |
| 16 | Softmax Layer |

*3.2. Loss Function (LF)*

The loss function (*LF*) is a task employed by models to assess their performance. Networks use automated learning to find rules and provide predictions for enormous amounts of data. The primary goal of the *LF* is to determine how much the real and anticipated values have changed. Throughout the model training process, the *LF* is adjusted regularly until a robust fitting value is obtained to reduce error.

We removed the final layer of the EfficientPNet model by introducing an output neuron to accomplish the categorization task for high-quality and distorted samples. For this reason, the hyperparameters of the framework were nominated using an empirical strategy. In our proposed approach, we have adopted the Adadelta optimizer during the model training phase, along with a learning score of 0.1. Moreover, we used twenty epochs for model training. The cross-entropy *LF* uses the Softmax function for classification tasks to assess the variance between calculated and real values. Calculating the cross-entropy *LF* is done as follows:

$$LF = \frac{1}{N}\sum\nolimits_{k=1}^{n} \log\left(\frac{e^{s_j}}{\sum_i e^{s_k}}\right) \tag{2}$$

Here, $N$ represents the total neurons, $s_k$ indicates the input vector, and $sj$ is the estimated label. The model permits the fine-tuning of only 20% of the entire framework parameters without adjusting the remaining 80%. A validation set was utilized to ensure the avoidance of model overfitting issues. Adaptive Moment Estimation [48] was adopted to compute the value of the learning rate against each parameter. This method works by storing the exponential decay of the previous gradient by adopting the impulse approach, as shown in Equations (3) and (4), respectively.

$$M_t = b1M_t - 1 + (1 - b1)G_t \tag{3}$$

$$V_t = b2V_t - 1 + (1 - b2)G^2{}_t \tag{4}$$

Here, $b1$, and $b2$ are constants with scores of 0.9, and 0.999, respectively, $G$ indicates the gradient score, and $M_t$ and $V_t$ represent the first-moment and second-moment vectors. The values of these two factors show the link between the updated and previous gradient values. Higher scores of these parameters show a close link between the previous and new

gradient values. Initially, the values of both moments are initialized to zero, which requires the bias correction factors $b1$ and $b2$ to avoid the 0 biases. Such biases can be removed by computing the bias-corrected Mt, as elaborated in Equations (5) and (6), respectively:

$$M_t = M_t - (bt1) \tag{5}$$

$$V_t = V_t - (bt2) \tag{6}$$

The optimization approach in our model uses Equation (7) to update the gradient value.

$$W_{t+1} = W_t - \eta / (V_t + \epsilon M)^{0.5} t \tag{7}$$

Here, $\epsilon$ is a constant, $\eta$ is a learning rate with a score of 0.00001, which is employed to avoid the denominator from becoming zero, and $W(t+1)$ shows the framework parameters at a given time $(t+1)$.

## 4. Experimental Results

This section briefly describes the dataset used to train and evaluate the classification results of the proposed technique for classifying various types of potato plant leaf diseases. In addition, it illustrates used performance measures. Finally, we carried out a huge comparison with various other models to show the effectiveness of our approach.

### 4.1. Dataset Acquisition

To check the recognition ability of our framework, a standard dataset called the PlantVillage repository [49] is utilized in this work. This data sample is free and available online for model simulation. The PlantVillage dataset is a large collection of plant leaf images with a total of 54,306 images. As the presented approach is associated with classifying plant leaf diseases only in potato crops, only a sample of the mentioned category was used for the performance evaluation. Table 2 demonstrates the list of categories included in the PlantVillage dataset. The reason for nominate this data sample for performance testing is that it comprises samples that vary in mass, structure, size, and orientation of both leaves and infected regions. Moreover, samples suffer from several distortions, including clutter, blur, intensity variations, and color variations. A few samples of this dataset are shown in Figure 3.

**Table 2.** List of categories included in the PlantVillage dataset without data augmentation.

| Class | Images in Dataset | Training Set | Test Set |
|---|---|---|---|
| Healthy Leaves | 600 | 480 | 120 |
| Early Blight | 1200 | 960 | 240 |
| Late Blight | 1200 | 960 | 240 |
| Total | 3000 | 2400 | 600 |

### 4.2. Data Augmentation

We used the PlantVillage dataset to obtain pictures of potato leaf diseases that we used to train, validate, and test the proposed DL model. The collection featured images of late blight, early blight, and healthy potato leaf conditions. The resolution of each image in the group was ($256 \times 256$) pixels. The images of healthy potato leaves portrayed leaves in a normal, healthy state. In contrast, the early and late blight photos illustrated the two stages of shattering potato leaf disease. For the three classes in the dataset, we used the indices 0, 1, and 2. The distribution of the dataset's total number of pictures among its many categories is shown in Table 2. In contrast to images of the other two groups of potato blight, the dataset contained far fewer pictures of healthy potato leaves. The dataset's photos were all randomly chosen to create a training and test set with an 80/20 ratio.

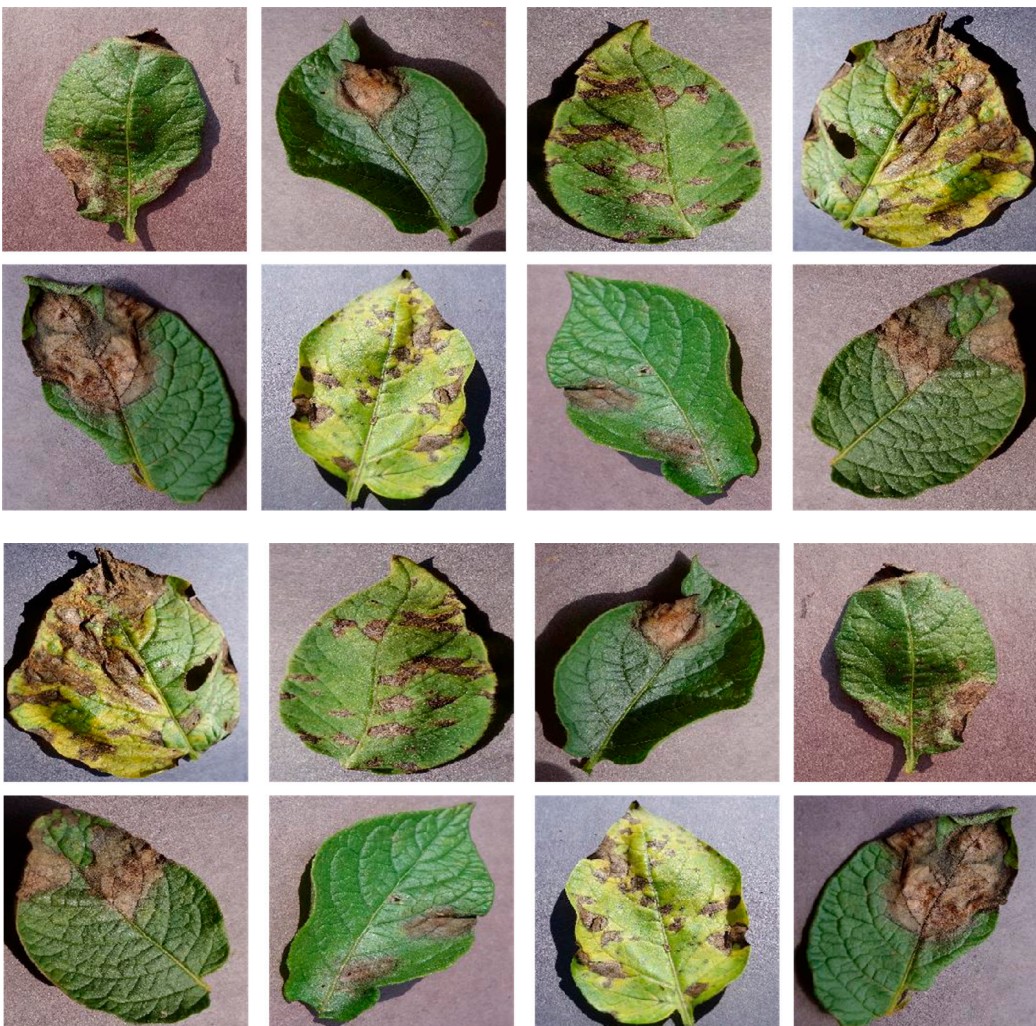

**Figure 3.** A visual figure of sample dataset.

By randomly picking ten healthy potato leaf photographs and making ten duplicates of each, we increased the quantity of healthy potato leaf images to balance the dataset. This procedure was repeated five times to balance the dataset in terms of photos of healthy potato leaves. Table 3 lists the total number of pictures in each class in the dataset after balancing. Originally, each category had 1200 images for early and late blight and 600 images of healthy potato leaves. After data augmentation, each category had 3600 images for early and late blight and 3600 images of healthy potato leaves.

**Table 3.** List of categories included in the PlantVillage dataset after data augmentation.

| Class | Images in Dataset | Training Set | Test Set |
|---|---|---|---|
| Healthy Leaves | 3600 | 2880 | 720 |
| Early Blight | 3600 | 2880 | 720 |
| Late Blight | 3600 | 2880 | 720 |
| Total | 10,800 | 8640 | 2160 |

We normalized the data and increased the size of the training set to train the model and ensure that it would not overfit. The photos were rotated between 20 and +20, sheared between 40 and +40, and moved by width and height within a range of 0.2 for augmentation. Figure 4 displays a visualization of the augmentation process.

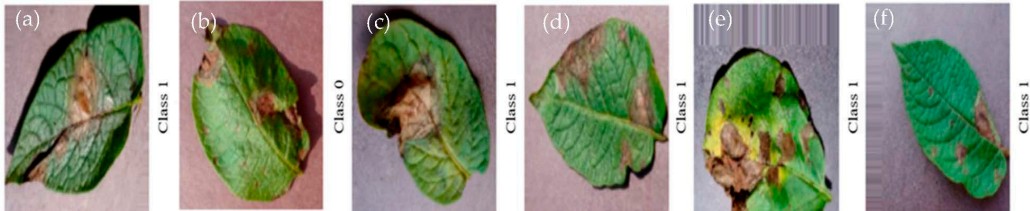

**Figure 4.** Visualized augmented images: (**a**) sheared, (**b**) rotated, (**c**) shifted, (**d**) vertically flipped, (**e**) horizontally flipped, (**f**) height-shifted.

### 4.3. Performance Metrics

To quantitatively estimate the categorization results of our approach for recognizing diseases of potato plant leaves, we used the standard measures of accuracy, F1 measure, precision (p), and recall (r). The mathematical formulation of the accuracy measure, p, r, and F1 is provided in Equations (8) to (11).

$$Accuracy = \frac{TP + TN}{TP + FP + TN + FN} \tag{8}$$

$$p = \frac{TP}{TP + FP} \tag{9}$$

$$r = \frac{TP}{TP + FN} \tag{10}$$

$$F1 = \frac{2 \times p \times r}{p + r} \tag{11}$$

### 4.4. Experimental Results

In the first phase of model evaluation, we tested the performance of the proposed strategy in terms of class-wise results to check how much well approach is able to recognize various types of potato plant leaf abnormalities. For this, we measured the performance of our approach using different performance metrics. The results are discussed below. In addition, the experimental results were verified by an expert who is currently working as a Plant Pathologist. Figure 5 shows the results for their training/validation loss and accuracy of the proposed model.

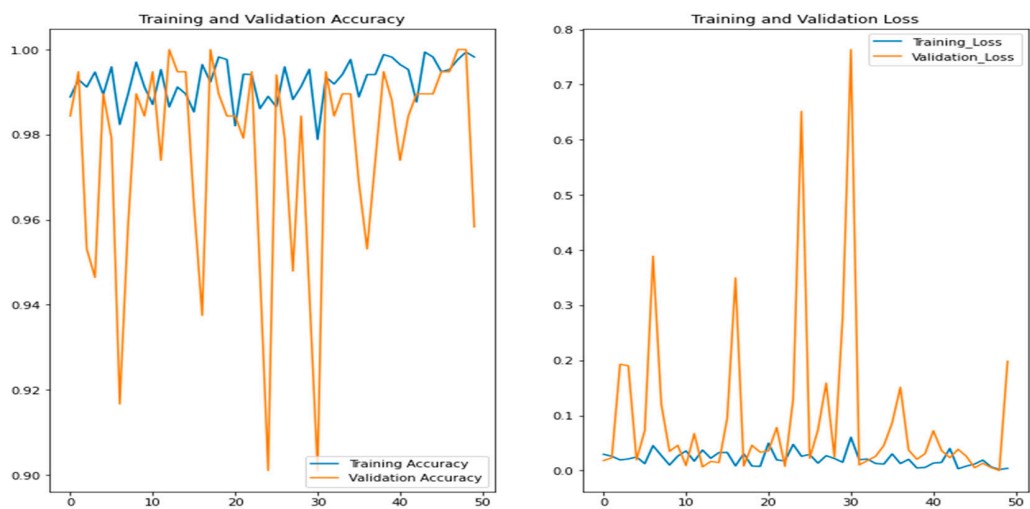

**Figure 5.** Graph of training and validation accuracy vs. loss for the proposed model.

First, the classification results of this approach are discussed in terms of precision and recall measures, as these are the standard way of elaborating model categorization results. The attained values are provided in Figure 6 for all three classes, showing healthy, early blight, and late blight, respectively. The scores attained in Figure 6 clearly indicate that our approach is able to effectively recognize all three classes in the employed dataset. For the precision metric our approach attained results of 98.26%, 98.03%, and 97.99% for healthy, early blight, and late blight, respectively, while for recall our solution showed values of 97.41%, 97.15%, and 97.10% for healthy, early blight, and late blight, respectively.

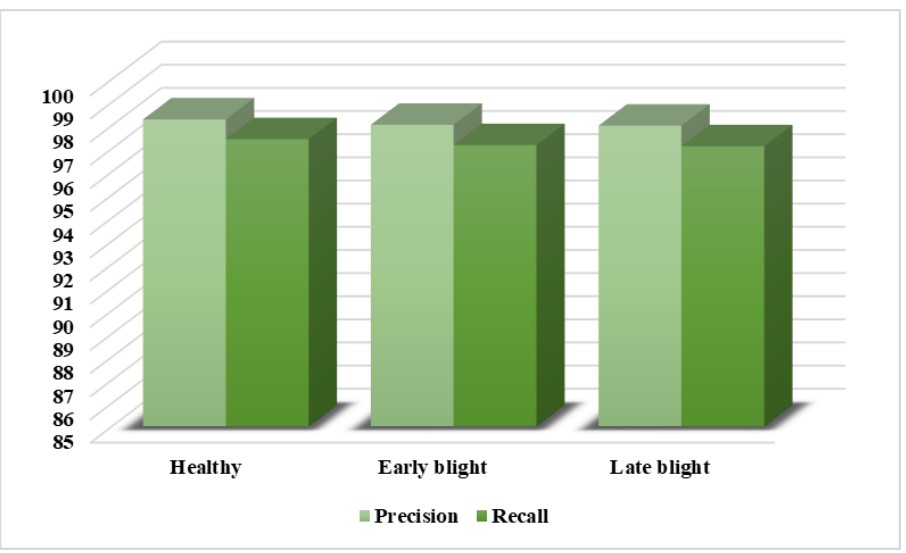

**Figure 6.** A graphical depiction of the attained precision and recall results.

Next, the model behavior is assessed from the perspective of the F1-score and error rate, as the precision and recall metrics are unable to fully capture the classification behavior of a model. This is because certain approaches are unable to attain a better value of recall for a high value of precision, and vice-versa. Hence, employing the F1-score measure can provide an overall performance assessment of a classification approach by employing both the precisions and recall measures. The attained results for all three classes of the employed dataset are provided in Figure 7. The suggested method reaches an average F1-score value of 97.65%, as depicted in Figure 7. Moreover, we the highest and lowest error scores are 2.46%, and 2.17%, for the late blight and healthy classes of potato plant leaves, respectively.

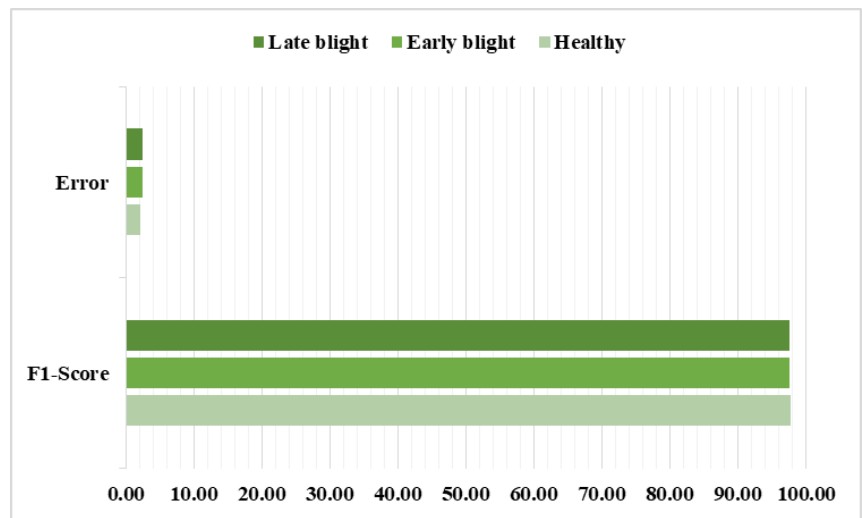

**Figure 7.** Graphical depiction of attained F1-score and error rate results.

Further, class-wise accuracy value was computed for all three groups of potato plant leaves; the obtained evaluation is shown with the help of box plots in Figure 8. Box plots are proficient in providing a thorough understanding of attained performance results by plotting the maximum, mean, and minimum values. The class-wise accuracy values shown in Figure 8 clearly prove the effectiveness of our approach for categorizing the infected areas of potato plant leaves. More descriptively, for the healthy, early, and late blight classes, the proposed solution acquires average values of 98.24%, 98.11%, and 98.01%, respectively.

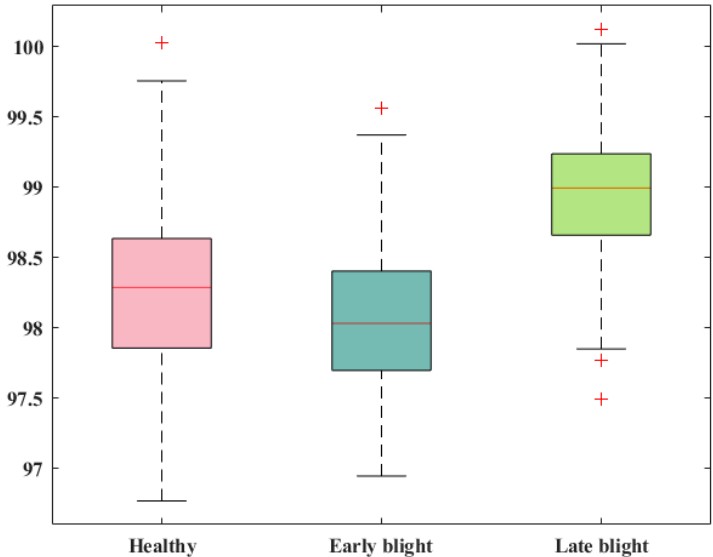

**Figure 8.** Graphical depiction of attained accuracy results.

Finally, we further depict the class-wise results of our approach by reporting the confusion matrix, which is a powerful plot for showing the recognition ability of a framework by reporting the values in terms of the true positive rate. The confusion matrix for our proposed strategy is shown in Figure 9, demonstrating that our model achieves better values on all three classes of potato plant leaves. Clearly, our approach attains an average TPR of %, which shows its better recall behavior. Moreover, we attain a minimum error of 97.22%, while the highest error rate of 1.61% is reported for the late and healthy blight classes, which could be due to the structural resemblance of the infected regions in these classes. However, both classes are highly differentiable.

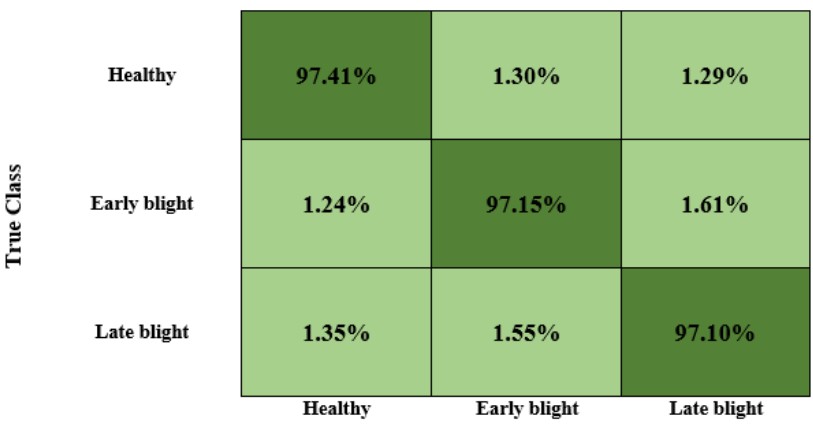

**Figure 9.** Attained results in the form of a confusion matrix.

All class-wise performance evaluations of the proposed solution with the help of the standard measures confirm the better recognition ability of our approach, which enables it to better classify all the samples in all three classes. The major reason for the improved classification behavior of our approach is due to the relevant and distinctive sample characteristics computation of our approach, which assists and enhances its recall rate and increases its classification performance.

### 4.5. Comparison with DL Models

In this section of the paper, a comparative analysis of the proposed work with other DL approaches is accomplished to show the efficacy of our work in comparison. For this purpose, a series of well-known DL frameworks, including VGG16 [50], VGG19 [51], MobileNet [52], ResNet50 [53], and DenseNet-101 [54], were nominated. We compared these DL architectures from the perspectives of model structure and performance by comparing the total number of model parameters and accuracy. The results of the evaluation are presented in Table 4. The values clearly depict our approach as being both effective and efficient in comparison to the other DL frameworks. Clearly, the presented work comprises the lowest number of model parameters, with 11 million. Comparatively, the VGG19 model is more expensive in terms of model structure, with a total of 1.96 million parameters. In terms of model accuracy, the lowest performance result is attained by ResNet50, with a score of 73.75%. The second lowest performance score is reached by MobileNet, at 78.84%. The DenseNet approach shows better performance outcomes, with an accuracy value of 93.93%; however, this approach is complex in terms of network structure, with a total of 40 million parameters. In comparison, our approach performs well with an accuracy score of 98.12% and has a total of 11 million model parameters. Clearly, the comparison of these approaches shows an average score of 83.92%, and is 98.12% for our model. Thus, we have achieved a performance gain of 14.20%, that clearly showing the efficacy of our model.

**Table 4.** Assessment of the suggested approach compared to other DL models.

| Sr No. | Model | Parameters (million) | Accuracy (%) |
|--------|-------|----------------------|--------------|
| 1. | VGG16 | 138 | 92.69 |
| 2. | VGG19 | 196 | 80.39 |
| 3. | MobileNet | 13 | 78.84 |
| 4. | ResNet50 | 23 | 73.75 |
| 5. | DenseNet | 40 | 93.93 |
| **6.** | **Proposed** | **11** | **98.12** |

The main cause of these better model classification results is that the other techniques are quite complex in terms of their model structure, which causes issues with model overfitting. Comparatively, our approach is lighter in structure and better able to tackle the overfitting issue. Moreover, our technique adopts the pixel and channel attention approach during the feature computation phase and introduces layers at the end of the network structure, which assists in better nominating the effective set of image characteristics and enhances the cataloguing score. Thus, it can be said that we have presented both an efficient and effective approach to recalling the various groups of potato plant leaf illnesses.

### 4.6. Proposed Approach in Comparison with the Latest Techniques

Next, we performed another experiment to check the potato plant disease classification results of our model against other new techniques from history. Numerous latest approaches [36,55–57] were nominated for this reason, and performance results in terms of classification results are evaluated. The attained performance comparison is shown in Table 5, from which it is quite clearly confirmed that our model is more robust for classifying the abnormalities of potato plants as compared to the other approaches shown in Figure 10.

**Table 5.** Comparison of the proposed approach with new methods.

| Sr. No | Reference | Accuracy (%) |
|--------|-----------|--------------|
| 1. | Chen et al. [55] | 97.73 |
| 2. | Barman et al. [56] | 96.98 |
| 3. | Mahum et al. [57] | 97.20 |
| 4. | Chakraborty et al. [36] | 97.89 |
| **5.** | **Proposed Technique** | **98.12** |

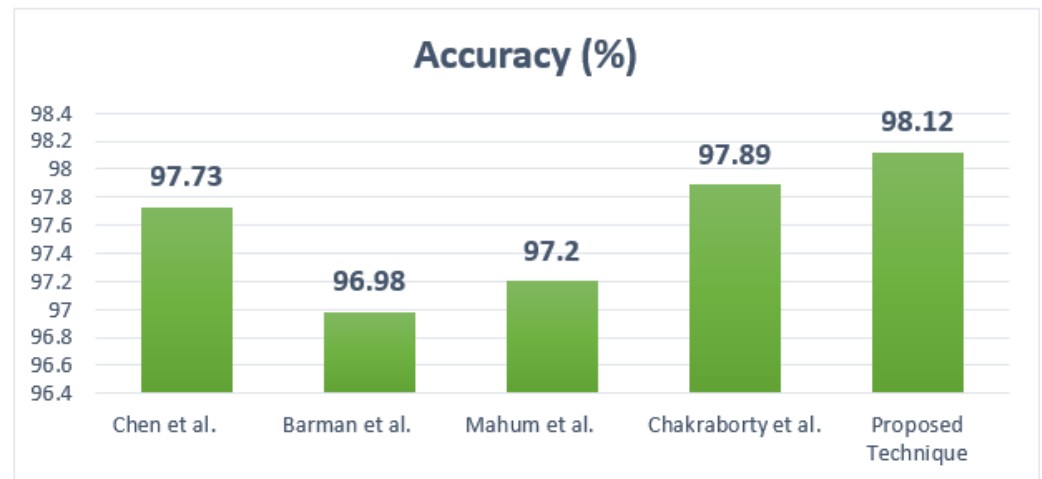

**Figure 10.** A comparison with the latest works developed by Chen et al. [55], Barman et al. [56], Mahum et al. [57] and Chakraborty et al. [36].

Chen et al. [55] used a DL approach called MobOca_Net to recognize different potato plant leaves by introducing pixel and channel-wise attention units in the base network. This approach attained an accuracy rate of 97.73%. Barman et al. [56] used a self-introduced CNN model to classify various infections found on the leaf areas of the potato crop, and achieved an accuracy of 96.98%. Another model, discussed in [57], used the concept of transfer learning to perform potato plant leaf diseases categorization, and attained a classification score of 97.20%, and the approach in [36] showed an accuracy value of 97.89%.

In comparison with these techniques, the proposed approach attains the highest accuracy rate at 98.12%. The compared techniques exhibit an average accuracy rate of 97.45%, compared to 98.12% for the presented strategy. Consequently, we have provided a performance gain of 0.67% in terms of the accuracy metric. The major cause of this effective performance result is that the approach in [55] is unable to tackle the distorted samples, while the technique in [56] lacks the ability to handle noisy data. On the other hand, the approaches in [36,57] suffer from issues with model overfitting. Comparatively, our approach is better able to handle these issues than existing approaches by presenting an effective model that adopts the pixel and channel AM in the feature computation phase and introduces dense layers at the end of the network structure, which results in nominating a reliable set of sample features even in the presence of various image distortions, thereby enhancing the classification score.

## 5. Conclusions

Farmers lose money and harvest due to potato plant diseases. Potato leaves are mostly affected by early and late blight. According to estimates, these illnesses are the cause of the majority of yield loss in potatoes. We divided photos of potato leaves into three categories: healthy leaves, late blight leaves, and early blight leaves. To recognize these classes, a solution called EfficientPNet is implemented in this paper. EfficientPNet is a DL approach that classifies various types of potato plant leaves. We improved the EfficientNet-v2 approach by adding the AM strategy and extra layers at the end of the

model structure. The presented EfficientPNet approach robustly extracts high-level signs of infected regions and associates them with related groups by employing an end-to-end learning mechanism. In addition, the AM strategy improves the recall power of the proposed solution by passing relevant information on noticeable attributes such as diseased areas of plant leaves. We accomplished rigorous experimentation on a complex data sample designated as PlantVillage to show the effectiveness of our framework, and proved through the attained performance scores that our model is proficient in recognizing potato diseases even from distorted images. As a future goal, we intend to develop another ensemble model by integrating explainable AI [58] and EfficientPNet DL architectures on other challenging datasets.

**Author Contributions:** Conceptualization, T.N., M.M.I., S.J. and A.H.; Data curation, T.N., M.M.I. and S.J.; Formal analysis, A.H. and M.A.; Funding acquisition, M.A.; Investigation, M.M.I.; Methodology, T.N. and M.A.; Project administration, M.A.; Resources, S.J. and A.H.; Software, S.J., A.H. and M.A.; Supervision, M.A.; Validation, T.N., S.J. and A.H.; Visualization, T.N. and M.A.; Writing—original draft, T.N., S.J., A.H. and M.A.; Writing—review and editing, M.M.I. and S.J. All authors have read and agreed to the published version of the manuscript.

**Funding:** The authors extend their appreciation to the Deanship of Scientific Research at Imam Mohammad Ibn Saud Islamic University (IMSIU) for funding and supporting this work through Research Partnership Program no. RP-21-07-11.

**Institutional Review Board Statement:** Not applicable.

**Informed Consent Statement:** Not applicable.

**Data Availability Statement:** A standard online dataset PlantVillage [49] is utilized in this paper to evaluate EfficientPNet model. It can be downloaded from https://data.mendeley.com/datasets/tywbtsjrjv/1 [accessed on 12 January 2023].

**Acknowledgments:** The authors extend their appreciation to the Deanship of Scientific Research at Imam Mohammad Ibn Saud Islamic University (IMSIU) for funding and supporting this work through Research Partnership Program no. RP-21-07-11. The authors would also like to thank our master student (Wasif Ali) for providing us with the code of EfficientPNet model from the Department of Computer Science, University of Engineering and Technology Taxila. This code was provided without data augmentation techniques.

**Conflicts of Interest:** The authors declare no conflict of interest.

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
