# Peer review of "EfficientPNet—An Optimized and Efficient Deep Learning Approach for Classifying Disease of Potato Plant Leaves"

_agriculture, doi:10.3390/agriculture13040841_

Round 1

Reviewer 1 Report

The manuscript is interesting as an improved technique for the recognition and classification of potato leaf diseases.

The topic is relevant for the journal, the methodology is detailed and easy to understand, the results section is well illustrated with relevant conclusions.

The introduction section is a little long in my opinion, the last part could rather be inserted in the discussion section to complete the full demonstration of all the advantages of the technique used, but it is only a suggestion.

Author Response

Original Manuscript ID:  ID: agriculture-2306237        

Original Article Title: EfficientPNet- An Optimized and Efficient Deep Learning Approach for Classifying Disease of Potato

To: Editor in Chief,

MDPI, agriculture

Re: Response to reviewers

Dear Editor,

Many thanks for insightful comments and suggestions of the referees. Thank you for allowing a resubmission of our manuscript, with an opportunity to address the reviewers’ comments.

We are uploading (a) our point-by-point response to the comments (below) (response to reviewers), (b) an updated manuscript with yellow highlighting indicating changes, and (c) a clean updated manuscript without highlights (PDF main document).

By following reviewers’ comments, we made substantial modifications in our paper to improve its clarity and readability. In our revised paper, we represent the improved manuscript such as:

(1) Revised Abstract, (2) Revised Introduction, (3) Results section, (4) Conclusion sections.

We have made the following modifications as desired by the reviewers:

Best regards,

Corresponding Author,

Dr. Sohail Jabbar (On behalf of authors),

Professor.

Reviewer 2 Report

The Efficient NetV2 technology proposed by the author utilizes MBConv and FMBConv blocks to achieve computational gain, and is used for potato leaf disease classification, improving accuracy and reducing the amount of parameters. It has certain application value and innovation.   But the author is requested to carefully examine the writing. For example, the numbering of reference 24 on line 158 is incomplete. In addition, lines 347 to 354 do not delete the redundant parts of the notes in the paper template.    There are many details to note in the article. Some places do not have punctuation marks, and whether the number of spaces between two words is correct, such as lines 163, 178, 180, 208, 254, and 331. It is recommended that the author unify the formula and formula symbols.

Author Response

(The authors gave the same response as above.)

Reviewer 3 Report

The authors have modified the existing EfficientNet V2 by adding new computation block (lines 240-290).

They followed set procedures. 

However, the present system is exclusive to distinguish between late and early blight diseases of potato only. The application must be validated under field conditions when conditions like look-alike diseases are encountered.

It is also not clear as to how many images were used by the authors and also whether authors used field photographs of varying degrees of resolution  for validation.

While the authors have just used early and late blight diseases, the title says diseases of plant leaves. Since, potato leaves are infected by several plant pathogens, the title may be modified as "........early and late blight of potato".

It appears that the paper is written by non-biologists and hence, the terminology is not appropriate. I suggest the paper may be reviewed by a Plant Pathologist so that the correct terms are used to communicate. For instance, diseases of plant leaves is normally written as foliar diseases. I have done some corrections but as it needs a good revision, I hope the authors will take care of it.

The manuscript may be modified and accepted for publication.

Author Response

(The authors gave the same response as above.)
